# Discovery of the First in Class 9-N-Berberine Derivative as Hypoglycemic Agent with Extra-Strong Action

**DOI:** 10.3390/pharmaceutics13122138

**Published:** 2021-12-12

**Authors:** Mikhail V. Khvostov, Elizaveta D. Gladkova, Sergey A. Borisov, Nataliya A. Zhukova, Mariya K. Marenina, Yuliya V. Meshkova, Olga A. Luzina, Tatijana G. Tolstikova, Nariman F. Salakhutdinov

**Affiliations:** N. N. Vorozhtsov Novosibirsk Institute of Organic Chemistry, Siberian Branch of the Russian Academy of Sciences, 9 Akademika Lavrentieva Ave., 630090 Novosibirsk, Russia; liza95@nioch.nsc.ru (E.D.G.); sergalborisov@mail.ru (S.A.B.); nazhukova.1958@yandex.ru (N.A.Z.); fominamk@gmail.com (M.K.M.); meshkova_29@mail.ru (Y.V.M.); luzina@nioch.nsc.ru (O.A.L.); tolstiktg@nioch.nsc.ru (T.G.T.); anvar@nioch.nsc.ru (N.F.S.)

**Keywords:** berberine derivative, hypoglycemic agents, hyperglycemia, in vivo, antidiabetic

## Abstract

Berberine is well known for its ability to reduce the blood glucose level, but its high effective dose and poor bioavailability limits its use. In this work we synthesized a new derivative of berberine, 9-(hexylamino)-2,3-methylenedioxy-10-methoxyprotoberberine chloride (**SHE-196**), and analyzed the profile of its hypoglycemic effects. Biological tests have shown that the substance has a very pronounced hypoglycemic activity due to increased insulin sensitivity after single and multiple dosing. In obese type 2 diabetes mellitus (T2DM) mice, it was characterized by improved glucose tolerance, decreased fasting insulin levels and sensitivity, decreased total body weight and interscapular fat mass, and increased interscapular brown fat activity. All these effects were also confirmed histologically, where a decrease in fatty degeneration of the liver, an improvement in the condition of the islets of Langerhans and a decrease in the size of fat droplets in brown adipose tissue were found. Our results indicate that 9-(hexylamino)-2,3-methylenedioxy-10-methoxyprotoberberine chloride could be the first in a new series of therapeutic agents for the treatment of diabetes mellitus.

## 1. Introduction

Berberine **1** (Figure 1) is an isoquinoline plant alkaloid. For centuries, berberine has been used in the traditional Chinese medicine to treat various diseases [1,2,3], one of its strongest features is reported to be its anti-diabetic activity. Despite the fact that the berberine **1** mechanism of antihyperglycemic effect is still unclear, its clinical use for type 2 diabetes mellitus has been reported since 1988. Basically, berberine inhibits α-glucosidase reducing the intestinal absorption of monosaccharides. In addition, berberine regulates the peroxisome proliferator-activated receptors and the expression of positive transcription elongation factor b in diabetic adipocytes [4]. It is also known to inhibit gluconeogenesis in the liver, modulate the FXR signaling pathway in the intestine [5,6], inhibit the LPS-induced TLR4/TNF-α activation, which enhances insulin receptor expression in the liver [7], activate AMPK pathways and inhibit the α-glucosidase enzyme [8]. In addition, berberine activates PPAR-γ and inhibits PTP1B activity [9]. Some other possible berberine mechanisms of action against obesity are modulation of the gut microbiota by increasing the amount of intestinal peptides (such as GLP-1, GLP-2 and peptide YY), decreasing the number of inhibitory gastric polypeptides by inhibiting the LXR alpha expression, cholesterol absorption and hepatic gluconeogenesis [10,11], suppression of adipocyte differentiation and proliferation by reducing galectin-3 levels [12], blocking adipogenesis by inhibiting CREB activity [13]. In other words, its anti-diabetic properties are likely to be of multitarget nature.

The hypoglycemic effect of berberine is very apparent, but the clinical dosage of berberine is relatively high (380 mg/kg) [14]. The main reason for that is the oral bioavailability of berberine being extremely low, not exceeding even 1% [15]. Due to its significant antibacterial activity, berberine cannot be used for the treatment of diabetes for an extended period of time.

The development of berberine as a glucose-lowering agent is now focused on improving its hypoglycemic activity by chemical modification. Sixteen various berberine derivatives **2** were synthesized and their antihyperglycemic activity was evaluated in a model of b-cell-membrane chromatography and a model of alloxan-induced diabetes in mice (Figure 1). The results indicated that two of them, namely compounds **2a**,**b**, exhibited antihyperglycemic activity. Their structure–activity relationships were discussed. All derivatives showed a hypoglycemic effect at a dose of 200 mg/kg [16].

The next step in identifying the most bioavailable berberine derivatives with hypoglycemic effect was to identify the corresponding effect of 8-oxoberberine **3** (Figure 1), a possible metabolite of berberine **1** in the body. [17]. The most effective hyperglycemic dose of compound **3** in a rat model of streptozotocin-induced diabetes was 100 mg/kg, as was reported by the authors.

Although many alternative positions can be modified, the modified position based on a certain biological activity is often specific, so for the hypoglycemic and hypolipidemic activity, the chemical modification sites are mostly focused on C-9 and C-10 [16,18,19,20,21,22,23,24]. A series of similar berberine-based derivatives were investigated to identify the more active and less toxic compounds, while certain aspects of SAR were identified from in vitro studies. One of the key modifications contributing to the hypoglycemic action was shown to be the modification at berberine position 9 to produce 9-O-acyl or alkyl derivatives [18].

In another study, several berberine derivatives were synthesized [25]. For each, the in vivo hypoglycemic action was investigated in a mouse model, resulting in the finding of a leading compound **5** (Figure 1) that demonstrated a profound hypoglycemic activity at a dose of 100 mg/kg [25].

The main goal of our research direction was to find a berberine derivative with the lowest effective dose. A novel berberine derivative 9-(hexylamino)-2,3-methylenedioxy-10-methoxyprotoberberine chloride **SHE-196** (Figure 2) was selected from the screening of a home-library of natural compounds derivatives in the OGTT (not published data) so the extended study of its hypoglycemic action in T2DM animals is the subject of present article. As an animal model of T2DM we used C57BL/6A^y^ (AY mice) mice which possess obesity, impaired glucose tolerance and concomitant non-alcoholic fatty liver disease [26]. In these animals, the OGTT and blood biochemical assays were performed. Additionally, their liver, pancreas and interscapular brown fat pad were subjected to histological examination.

## 2. Materials and Methods

### 2.1. Chemistry

^1^H and ^13^C NMR spectra were acquired on Bruker spectrometers AV-400 at 400.13 MHz (^1^H) and 100.61 MHz (^13^C). Spectra were recorded in deuterated dimethyl sulfoxide (DMSO-d6); residual DMSO-d6 was used as a standard [δ(DMSO-d6) 2.50, δ(DMSO-d6) 39.51 ppm] for measuring for measuting chemical shifts δ in parts per million (ppm), J was measured in Hertz. The structure of the product was determined by means of ^1^H and ^13^C NMR spectra and confirmed by MS spectrum. For the column chromatography, silica gel (60–200 mesh, Macherey–Nagel) was used. Mass-spectra was recorded at Bruker micrOTOF*_Q_* using electrospray ionization (ESI). Spectral and analytical measurements were carried out at the Multi-Access Chemical Service Center of Siberian Branch of Russian Academy of Sciences (SB RAS). Berberine chloride hydrate was purchased from TCI Co. (Tokyo, Japan) and used after drying, hexylamine was purchased from Acros Organics (China) and used without additional purification.

#### 9-(hexylamino)-2,3-methylenedioxy-10-methoxyprotoberberine chloride (SHE-196)

Berberine chloride **1** was dried in oven for 2 h at 95 °C. Dried berberine chloride (2 g, 5.4 mmol) was treated with hexylamine (2.5 mL, 19.0 mmol). The reaction mixture was heated to 125 °C and stirred for 4 h. Then the acetone was added, and the resulting burgundy precipitate was filtered. The precipitate was purified by column chromatography (eluent—CHCl_3_:MeOH (100:4) to give compound **SHE-196** (1.23 g, 52%) as a burgundy solid. ^1^H NMR (400 MHz, DMSO-d_6_): δ 10.10 (s, 1H, H-8), 8.68 (s, 1H, H-13), 7.88 (d, J = 8.7 Hz, 1H, H-9*), 7.75 (s, 1H, H-1), 7.46 (d, J = 8.7 Hz, 1H, H-10*), 7.06 (s, 1H, H-4), 6.40 (t, J = 6.07 Hz, 1H, NH), 6.15 (s, 2H, OCH_2_O), 4.79 (t, J = 6.1 Hz, 2H, H-6), 3.59 (s, 3H, OCH_3_), 3.54–3.60 (m, 2H, NH*CH*_2_), 3.18 (t, J = 6.1 Hz, 2H, H-5), 1.57–1.64 (m, 2H, NHCH_2_*CH*_2_), 1.26–1.35 (m, 6H, NHCH_2_CH_2_*CH*_2_*CH*_2_*CH*_2_), 0.85 (t, J = 6.9 Hz, 3H, CH_2_*CH_3_*). ^13^C NMR (100 MHz, DMSO-d_6_): δ 149.45, 147.58, 146.54, 137.24, 135.85, 133.00, (C-2, C-3, C-4a, C-10, C-12a, C-13a), 146.38 (C-8), 130.20, 120.57, 116.83 (C-8a, C-9, C-13b), 124.45 (C-13), 119.62 (C-12), 115.91 (C-11), 108.40 (C-4), 105.17 (C-1), 101.95 (OCH_2_O), 56.92 (OCH_3_), 54.74 (C-6), 47.24 (NHCH_2_), 31.03, 30.43, 26.62, 26.00, 22.04 (NHCH_2_*CH*_2_*CH*_2_*CH*_2_*CH*_2_, C-5), 13.88 (CH_2_*CH_3_*). MS (ESI): *m*/*z* (M^+^) calcd for C_25_H_29_N_2_O_3_, 405.217 found: 405.217.

### 2.2. Biological Experiments

Statistical analysis was performed by the Mann–Whitney *U* test. Data are shown as mean ± SEM. Data with *p* < 0.05 were considered statistically significant.

#### 2.2.1. Animals

The study involved male C57BL/6, CD-1 mice weighing 22–25 g and male AY mice weighing 28–32 g. Animals were obtained from the SPF vivarium of the Institute of Cytology and Genetics SB RAS. The animals were housed in polycarbonate cages with ad libitum access to water and feed. In the rooms of vivaria humidity, temperature and 12/12 h light-and-dark cycle were controlled. All manipulations with animals were carried out in strict accordance with the laws of the Russian Federation, a decree of the Ministry of Health of the Russian Federation no. 199n of 4 January 2016, and Directive 2010/63/EU of the European Parliament and of the Council of the European Union of 22 September 2010 on the protection of animals used for scientific purposes. The protocol of the animal experiment was approved by the Ethics Committee of N.N. Vorozhtsov Institute of Organic Chemistry SB RAS (protocol no. P-02-02.2021-14).

#### 2.2.2. The OGTT

The test was performed on mice (C57BL/6 or AY, *n* = 6 in each group) after a 12 h fasting. Oral glucose loading (2.5 g/kg) was done in all groups of mice. Prior to dissolution in water, **SHE-196** was mixed with two drops of Tween 80 and administered at a dose of 5, 10 and 15 mg/kg. Vildagliptin (VLD) tablets (Galvus, Novartis Farmaceutica SA, Barcelona, Spain) were dissolved in water and used as the positive control at a dose of 10 mg/kg in the OGTT in C57BL/6 mice. In the case of the AY mice experiment a water solution of metformin (MF, CAS 1115-70-4 Acros Organics, Geel, Belgium) at a dose of 250 mg/kg was used as the positive control according to the experimental design 2.2.3. All compounds were introduced 30 min prior to the glucose load by oral gavage. Blood samples were obtained from tail incision before dosing (time 0) and at 30, 60, 90, and 120 min after the glucose load. Blood glucose concentration was evaluated with a ONE TOUCH Select blood glucose meter (LIFESCAN Inc., Milpitas, CA, USA). The area under the glycemic curve (AUC) was calculated using Tai’s model [27].

#### 2.2.3. The Design of the Experiment on AY Mice

In order to facilitate the body weight gain mice were fed standard chow plus lard and cookies ad libitum for 30 days. Animals with body weight more than 35 g were selected for the experiment and divided in the following groups: (1) AY mice (*n* = 6) + vehicle (water + 2 drops of Tween 80), (2) AY mice (*n* = 6) + **SHE-196** 15 mg/kg, (3) AY mice (*n* = 6) + MF 250 mg/kg, and (4) C57BL/6 mice (*n* = 6) + vehicle (water + 2 drops of Tween 80). The animals were on the same diet till the end of the experiment. All compounds were given once a day by oral gavage. OGTT was performed on the 14th day of the experiment. On day 22 animals were decapitated and blood was drawn for the biochemical assay and insulin measurement. Liver, interscapular brown fat and pancreas were excised for histological evaluation.

#### 2.2.4. Insulin Blood Concentration Evaluation after Single Administration

CD-1 mice (*n* = 5), fasted for 4 h, were treated orally (oral gavage) by **SHE-196** at a single dose of 15 mg/kg or by water with 2 drops of Tween 80. The experiment was conducted on the same animals on different days. Blood samples of 0.1 mL were withdrawn from animals’ tail prior to introduction (0 point) and 30, 45 and 60 min after the introduction. After coagulation (not less than 30 min) blood samples were centrifuged for 15 min at 1640× *g* for 15 min for serum separation. Serum samples were frozen for further insulin ELISA evaluation.

#### 2.2.5. Insulin ELISA Examination

To analyze insulin serum concentration, the rat/mouse insulin ELISA kit (Cat. # EZRMI-13K, Millipore, Merck KGaA, Darmstadt, Germany) was used. Sample preparation and all procedures were done according to the manufacturers guide. Multiscan Ascent (Thermo Labsystems, Helsinki, Finland) photometer was used for analysis.

#### 2.2.6. Biochemical Assays

After 22 days of treatment, mice were decapitated, blood was collected and serum was separated by centrifugation at 1640× *g* for 15 min. Serum total cholesterol, triglycerides, total protein, alkaline phosphatase, alanine aminotransferase and lactate levels were quantified in all groups using standard diagnostic kits (Vector-Best, Novosibirsk, Russia) and a Stat Fax 3300 spectrophotometer (Awareness Technology Inc., Palm City, FL, USA).

#### 2.2.7. Toxicology Study

In order to find the mean lethal dose (LD_50_) **SHE-196** was introduced by oral gavage to CD-1 mice (*n* = 6) at a single dose of 30, 50 and 100 mg/kg. Animals’ mortality was evaluated for the next 10 days. LD_50_ was calculated by the Finney’s Probit analysis [28]. During the first 6 h after the administration blood glucose was measured with a ONE TOUCH Select blood glucose meter (LIFESCAN Inc., Milpitas, CA, USA). Blood samples were obtained from tail incision.

#### 2.2.8. Histological Examination

Liver, interscapular brown fat and pancreas were fixed in 10% neutral buffered formalin for 7 days, then standard dehydration in ascending ethanol concentrations and xylene was carried out. All samples were embedded in paraffin on an AP 280 workstation using Histoplast (Thermo Fisher Scientific, Waltham, MA, USA) with a melting point of 58 °C. Tissues were sliced with a thickness of 4.5 μm on a rotational microtome NM 335E with disposable interchangeable blades. The slices were stained with periodic acid–Schiff, hematoxylin and eosin, and orange G and examined under a light microscope at a magnification of ×100–200.

## 3. Results

### 3.1. Synthesis

The compound **SHE-196** was synthesized from berberine chloride in one step by its reaction with 1-hexylamine. Compound **SHE-196** was purified by column chromatography on SiO_2_ (eluent—4% methanol in chloroform) and was obtained with yield 52%.

### 3.2. The OGTT in C57BL/6 Mice

Prior to starting experiments in AY mice, we performed the OGTT in C57BL/6 mice. As a positive control vildagliptin was used. It was found that **SHE-196** possesses hypoglycemic action at a dose range from 5 to 15 mg/kg. The dose of 15 mg/kg was the most powerful one, even more effective than VLD ata 10 mg/ kg dose (dose is equal to **SHE-196′s** 15 mg/kg due to its larger molar mass) (Figure 3).

### 3.3. AY Mice Experiments

At the beginning of the AY mice treatment fasted blood glucose levels and animals’ body mass were evaluated. Thereafter, body mass was recorded once a week (Figure 4) and blood glucose levels after two and three weeks of treatment (Figure 5). The **SHE-196** dose that was used (15 mg/kg), was the most effective one from the early performed OGTT. From the obtained data it is clearly evident that treatment with **SHE-196** and MF, which was the positive control, during 3 weeks resulted in a dramatical reduction in fasted glucose levels and body mass.

Thus, after two weeks of treatment, MF and **SHE-196** significantly improved AY mice’s glucose tolerance and it is worth mentioning that **SHE-196** did it more prominently than MF (Figure 6 and Figure 7).

After three weeks of administration the experiment was stopped and animals were decapitated in order to collect blood for the biochemical examination and insulin measurement, also gonadal and interscapular fat masses were examined.

Mice treated by MF and **SHE-196** showed reduction in both gonadal and total interscapular fat pads masses. Brown fat mass in both of these groups were also decreased (Table 1).

Fasted insulin level examination revealed a dramatically low concentration in the animals, treated with SHE-196, compared to untreated AY mice (Table 2).

During a biochemical blood assay, it was found that lactate levels in mice treated by MF and **SHE-196** were significantly decreased. Triglycerides and alkaline phosphatase levels in **SHE-196** group has some tendency to decrease in comparison to untreated AY mice. Total cholesterol, total protein and alanine aminotransferase levels were the same as in untreated AY mice (Table 3).

### 3.4. Insulin Blood Concentration Evaluation after Single Administration

In addition to AY mice three weeks treatment we performed the examination of insulin blood concentration after **SHE-196** single oral administration. This experiment was conducted in CD-1 mice. The results showed that **SHE-196** at a dose of 15 mg/kg after single oral introduction is able to increase tissues’ sensitivity to insulin which is reflected in the insulin blood concentration decrease (Figure 8).

### 3.5. Toxicology Study

The toxicology study revealed that the LD_50_ of **SHE-196** is equal to 63 mg/kg but it should be emphasized that the mortality was due to the very prominent blood glucose decrease: <1.1 mmol/L 6 h after the single oral introduction.

### 3.6. Hystology

In the group of control C57BL/6 mice, the liver architectonics was preserved, bile capillaries, veins and arteries had a typical structure, and signs of pathological infiltration, necrosis and fibrosis were absent. In brown adipose tissue, adipocytes containing mainly small drops of fat were detected. In the pancreas, a slight plethora of capillaries was detected, the state of the exocrine and endocrine regions was not changed (Figure 9A).

In the liver of the control group’s AY mice, dystrophic changes in hepatocytes of the periportal zones were observed in the form of the perinuclear zone’s cytoplasm devastation and small vesicular lipid infiltration. In all animals heterogeneity of nuclei and size of hepatocytes was shown, activated Kupffer cells in sinusoids (an increase in their size and number) were odserved. The total large-drop fatty infiltration of the liver was revealed. Brown adipose tissue consisted of fat cells, densely braided by hemocapillaries and narrow layers of connective tissue. Adipocytes contained a large number of fat droplets, predominantly large, which merged with each other, leading to the formation of fatty cysts. In the pancreas, venous congestion was detected. The exocrine part of the gland had a typical structure. All AY mice, in comparison with the C57Bl/6 group, had hyperplasia of the islets of Langerhans of varying severity (Figure 9B).

Normalization of the liver structure was observed in AY mice treated with metformin. Signs of pathological infiltration, necrosis and fibrosis were not identified. It should be noted that the fat content in brown adipose tissue decreased slightly. In the pancreas, against the background of venous plethora, insignificant signs of islet hyperplasia remained (Figure 9C).

AY mice treated with **SHE-196** showed positive dynamics of pathological changes as compared with the control AY mice group. In hepatocytes, small vesicular lipid infiltration was detected in only one animal from the entire group. There was a slight heterogeneity of the hepatocyte nuclei, which may be a sign of an increase in the regenerative activity of the liver. A large number of enlarged Kupffer cells were noted in the sinusoids. Signs of structural rearrangement of liver tissue were not found. In the exocrine and endocrine pancreas, there were no signs of plethora, structural rearrangement, and hyperplasia. Small drops of fat were observed in adipocytes of the brown adipose tissue (Figure 9D).

White adipose tissue had a typical structure in all experimental groups.

## 4. Discussion

A novel berberine derivative **SHE-196** was selected from screening of a home-library of natural compounds’ derivatives in the OGTT. **SHE-196** was obtained by us in one step from berberine chloride and 1-hexylamine using an adapted procedure [29] for the synthesis of N-substituted berberine derivatives. Some N-alkyl-substituted derivatives are described in the literature, namely N-propyl, pentyl (etc.), for which only antibacterial properties against *Staphylococcus aureus* are described [30]. To the best of our knowledge, neither hypoglycemic nor hypolipidemic properties for this type of compounds have been previously described.

Our in vivo study began with an evaluation of the dose-dependent hypoglycemic activity of a new berberine derivative **SHE-196**. As a result of the OGTT, it was found that this substance has the ability to reduce the concentration of blood glucose in the dose range from 5 to 15 mg/kg. At the same time, at doses of 10 and 15 mg/kg, it was more effective than the positive control VLD at a dose of 10 mg/kg. VLD as a DPP4 inhibitor increases insulin secretion [31] and used as a positive control in screening by OGTT since its dose rate is relatively similar (due to molecular weights) to the new chemicals tested. The next step was to study the hypoglycemic effect of **SHE-196** in an animal model of T2DM-AY mice. These mice are characterised by obesity, hyperinsulinemia and hyperglycemia which develop with age. All these metabolic changes are the result of violation in the melanocortin pathway of appetite regulation due to mutation at the agouti locus which cause uncontrolled expression of the Agouti gene in all tissues. Gene’s product, Agouti protein, is the antagonist of melanocortin receptors and thereby its increased production decrease activity of these receptors [32]. In this experiment, we used metformin at a dose of 250 mg/kg as a positive control. Generally MF considered as a starting therapy for patients with T2DM. Its hypoglycemic effect is manifested through several mechanisms of action: inhibition of gluconeogenesis in the liver and enchancement of insulin sensivity [33]. According to the literature, its effective dose ranges from 200 to 250 mg/kg [34], so it was inappropriate to reduce it to the **SHE-196** level, since it was important for us to see the positive effects in the control group. The administration of **SHE-196** at a dose of 15 mg/kg for three weeks significantly affected the condition of the animals. All the main markers of the metabolic syndrome inherent in these mice were normalized: reduced body weight, decreased fasting glucose and insulin concentrations, and improved glucose tolerance. The decrease in body weight occurred, among other things, due to a decrease in the mass of gonadal and interscapular fat, as well as brown fat, which indicates an increase in the activity of the latter [27]. In the biochemical study, a decrease in the level of lactate in the blood of animals treated with **SHE-196**, as well as metformin, was found. This is an additional confirmation of the improvement in the sensitivity of mouse tissues (liver, muscle, fat) to insulin [35]. The biochemical markers reflecting liver function did not differ from those in negative control animals, which indicates the absence of hepatotoxicity. The level of TG in the blood of mice treated with **SHE-196** tends to decrease, which can be regarded as an acceleration of their oxidation in adipose tissue, including brown. Most likely, with a longer administration of the studied substances, this parameter would decrease even more. The fact that one (possibly the main) of the mechanisms of the hypoglycemic action of **SHE-196** is a pronounced increase in tissue sensitivity to insulin is evidenced by the results of an experiment on CD-1 mice. Its single administration at a dose of 15 mg/kg leads to a glucose-independent gradual decrease in the level of insulin in the blood, which most likely explains its hypoglycemic effect in the OGTT performed before the experiment with AY mice. An increase in the sensitivity of the liver and skeletal muscles to insulin due to an increase in the number of its receptors in these tissues was previously demonstrated for berberine itself at doses of 75 and 150 mg/kg administered orally for three weeks [36], but this does not explain such a rapid decrease in blood insulin concentration after a single gavage of **SHE-196** and requires further research.

It can be assumed that the mechanism of action of **SHE-196** somewhat differs from the initial berberine, since for the latter, in addition to a decrease in body weight, fasting glucose and an increase in glucose tolerance in OGTT, a decrease in the level of TC in the blood and no effect on the level of insulin were noted [14]. However, according to the obtained results we assume that **SHE-196** should keep some main berberin’s mechanisms of action such AMPK and MAPK activation and also be a multitarget molecule [37].

In the literature there are several studies dealing with different berberine derivatives and studing their gypoclycemic action. For example, introduction of free carboxyl groups at tetrahydroberberine moity at N-7 or C-6 and acetic group attachement at C-6 position of protoberberine exhibited glucose lowering effect but only in vitro. In another study several structural modifications at 9-O position and carbohydrate modified berberine derivatives showed good efficacy in in vitro gypoglycemic evaluations [38].Closer to our study is the work by Bian et al. where 7-N-alkyl derivative of berberine was shown to be the most effective in lowering blood glucose in alloxan-induced diabetes in rats but at a dose of 200 mg/kg [16].

The oral LD_50_ for the **SHE-196** was 63 mg/kg, which, of course, is not high. However, all observed mortality in animals was directly related to the main pharmacological action of **SHE-196**. When measuring the level of glucose in the blood of all animals that later died, there was a marked decrease in this indicator to values below 1.1 mmol/L (the lower limit of the glucometer’s detection). Thus, the animals died from hypoglycemic shock. Currently, there is not much diversity among therapeutic agents for increasing tissue sensitivity to insulin and lowering blood glucose levels. Drugs with such an effect are metformin and PPARγ agonists, the effect of all other drugs used is usually associated with an increase in insulin secretion (sulfonylurea derivatives, DPP4 inhibitors), a decrease in glucose absorption in the gastrointestinal tract (acarbose), or with its increased excretion by the kidneys (SGLT2 inhibitors) [39]. Therefore, it is extremely interesting to search for new agents that increase the sensitivity of tissues to insulin, since this is one of the most important disorders of carbohydrate metabolism in T2DM [39]. **SHE-196** may be the progenitor of a new line of such compounds, which undoubtedly requires further research both on itself and on new berberine derivatives of a similar structure in order to obtain effective agents, but with a wide therapeutic range.

## 5. Conclusions

In this work, we synthesized a novel berberine derivative—9-(hexylamino)-2,3-methylenedioxy-10-methoxyprotoberberine chloride (**SHE-196**), and analyzed the profile of its effects relevant to hypoglycemic action. The biological experiments showed that the studied substance possesses a very prominent hypoglycemic activity due to the increased insulin sensitivity after single and multiple dosing. In mice with T2DM and obesity it improved glucose tolerance, reduced fasted insulin level and insulin sensitivity, decreased total body and interscapular fat mass and also increased the activity of interscapular brown fat. All these effects were also confirmed histologically, where a decrease in fatty degeneration of the liver, an improvement in the condition of the islets of Langerhans and a decrease in the size of fat droplets in brown adipose tissue were found. Thus, our results mean that 9-(hexylamino)-2,3-methylenedioxy-10-methoxyprotoberberine chloride can be the first one in a novel line of therapeutic agents for diabetes mellitus treatment.

## Figures and Tables

**Figure 1 pharmaceutics-13-02138-f001:**
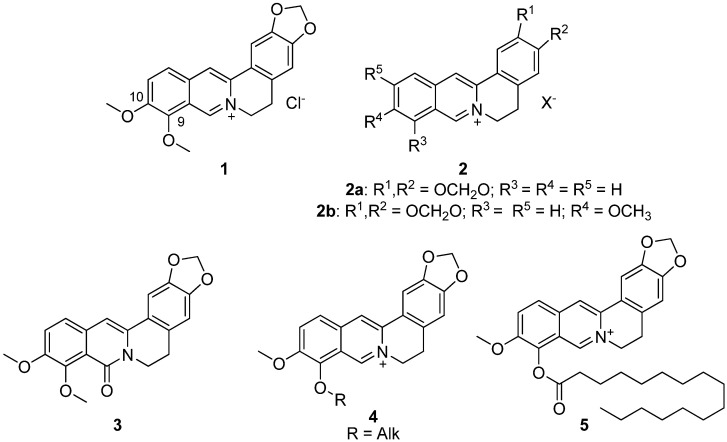
Structures of berberine **1** and its derivatives **2**–**5**, which possessed hypoglycemic activity.

**Figure 2 pharmaceutics-13-02138-f002:**
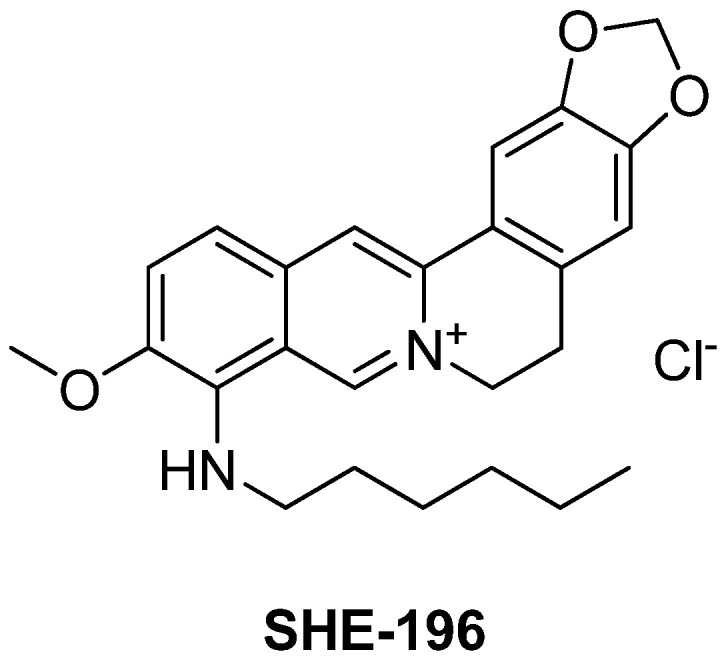
Structure of 9-(hexylamino)-2,3-methylenedioxy-10-methoxyprotoberberine chloride **SHE-196**.

**Figure 3 pharmaceutics-13-02138-f003:**
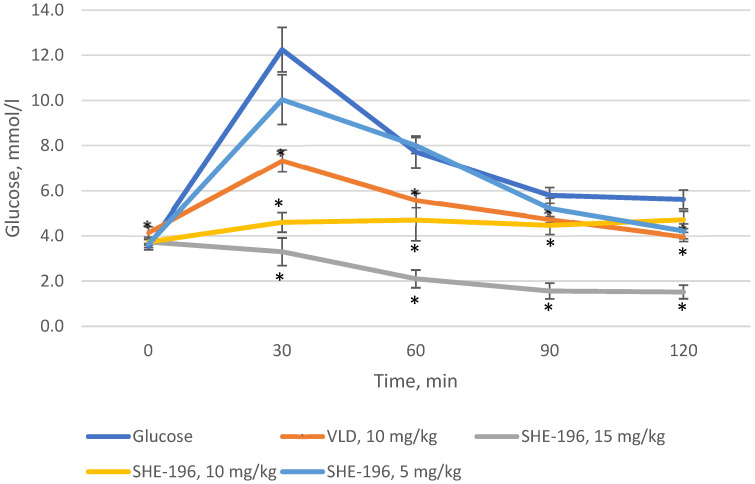
OGTT in C57BL/6 mice prior to the experiment with AY mice. * *p* < 0.05 compared to Glucose group.

**Figure 4 pharmaceutics-13-02138-f004:**
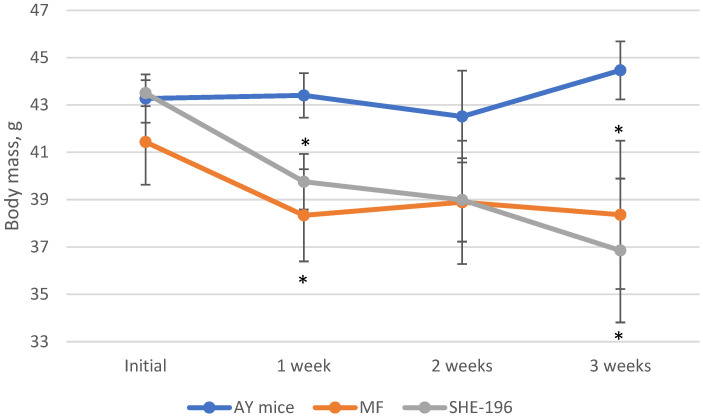
Body mass evaluation during 3-week treatment of AY mice by **SHE-196** at a dose of 15 mg/kg. MF was introduced at a dose of 250 mg/kg. * *p* < 0.05 compared to AY mice.

**Figure 5 pharmaceutics-13-02138-f005:**
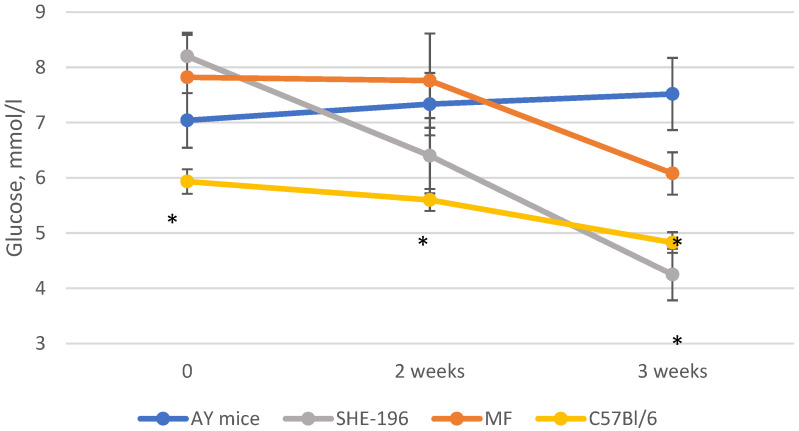
Fasted glucose levels during the experiment. * *p* < 0.05 compared to AY mice.

**Figure 6 pharmaceutics-13-02138-f006:**
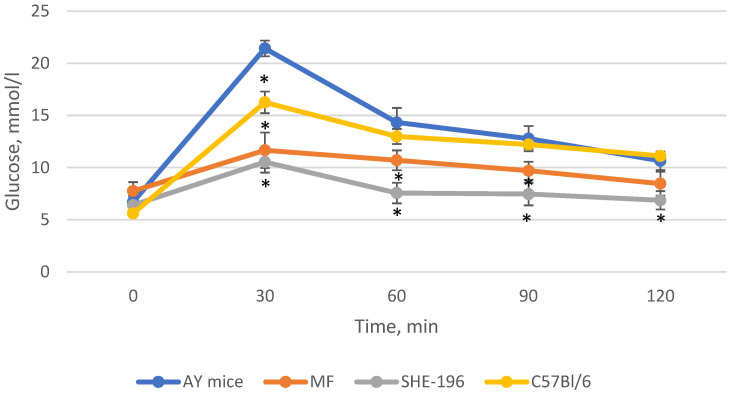
The results of the OGTT performed after 2 weeks of AY mice treatmentby **SHE-196** at a dose of 15 mg/kg. MF was introduced at a dose of 250 mg/kg. * *p* < 0.05 compared to AY mice.

**Figure 7 pharmaceutics-13-02138-f007:**
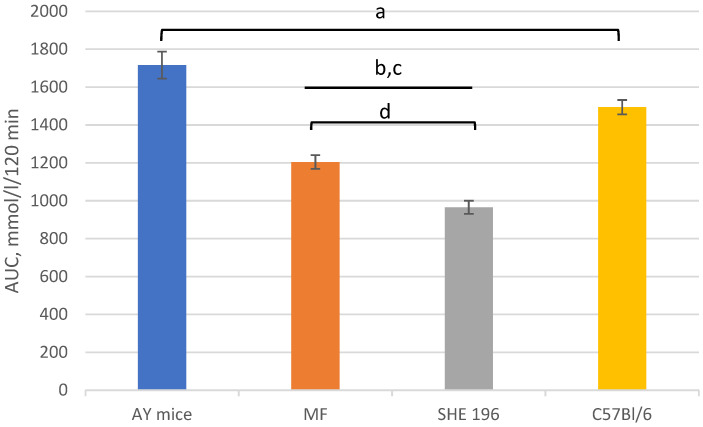
Area under the glycemic curve calculated according to the OGTT data after 2 weeks of AY mice treatment by **SHE-196** at a dose of 15 mg/kg and MF at a dose of 250 mg/kg. a, b, c: *p* < 0.05 compared to AY mice; d: *p* < 0.05 compared to MF.

**Figure 8 pharmaceutics-13-02138-f008:**
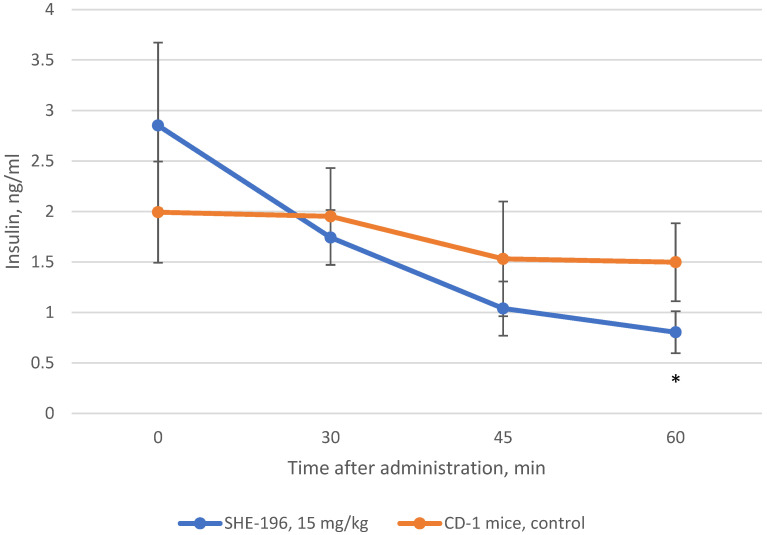
Change in insulin blood concentration after single oral introduction of **SHE-196** at a dose of 15 mg/kg in CD-1 mice. * *p* < 0.05 compared to control animals.

**Figure 9 pharmaceutics-13-02138-f009:**
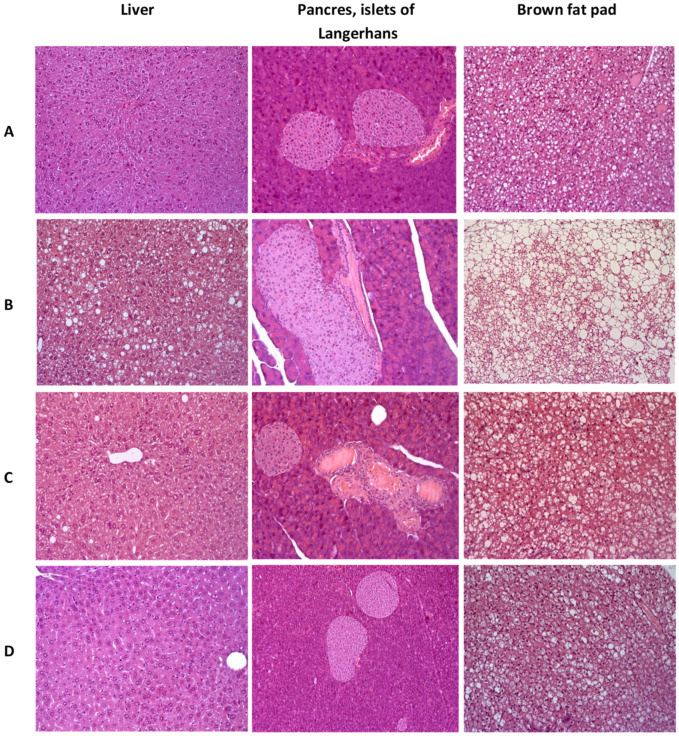
Histologial evaluation of liver, pancreas, and brow fat pad in mice after 3 weeks of experiment. (**A**)—C57Bl/6 (healthy control), (**B**)—AY mice (untreated), (**C**)—AY mice treated by metformin at a dose of 250 mg/kg, (**D**)—AY mice treated by **SHE-196** at a dose of 15 mg/kg. Hematoxylin and eosin staining, ×200.

**Table 1 pharmaceutics-13-02138-t001:** Mass of gonadal fat pads, interscapular fat pads and brown fat in AY mice treated for three weeks by **SHE-196** at a dose of 15 mg/kg and MF at a dose of 250 mg/kg.

Group	Gonadal Fat Pad, g	Interscapular Fat Pad, g	Interscapular Brown Fat, g
AY mice	2.51 ± 0.18	1.21 ± 0.02	0.21 ± 0.02
MF	1.91 ± 0.17	0.87 ± 0.09 *	0.13 ± 0.02 *
SHE-196	2.14 ± 0.27	0.92 ± 0.19 *	0.12 ± 0.03 *
C57Bl/6	0.29 ± 0.07	-	0.03 ± 0.003 *

* *p* < 0.05 compared to AY mice.

**Table 2 pharmaceutics-13-02138-t002:** Fasted insulin levels in AY mice treated for three weeks by **SHE-196** at a dose of 15 mg/kg and MF at a dose of 250 mg/kg.

Group	Insulin, ng/mL
AY mice	6.08 ± 1.01
MF	2.23 ± 0.51 *
SHE-196	1.75 ± 0.54 *
C57Bl/6	0.45 ± 0.11 *

* *p* < 0.05 compared to AY mice.

**Table 3 pharmaceutics-13-02138-t003:** Blood biochemical parameters of AY mice treated for three weeks by **SHE-196** at a dose of 15 mg/kg and MF at a dose of 250 mg/kg. TC—total cholesterol, TG—triglycerides, TP—total protein, APH—alkaline phosphatase, ALT—alanine aminotransferase.

Group	TC, mmol/L	TG, mmol/L	TP, g/L	Lactate, mmol/L	APH, U/L	ALT, U/L
AY mice	2.54 ± 0.11	0.28 ± 0.03	63.49 ± 3.65	8.41 ± 0.54	98.50 ± 5.94	45.38 ± 3.36
MF	2.52 ± 0.14	0.25 ± 0.02	59.70 ± 3.64	5.88 ± 0.41 *	94.60 ± 94.68	46.20 ± 2.65
SHE-196	2.65 ± 0.28	0.21 ± 0.01	59.93 ± 3.49	5.15 ± 0.24 *	85.50 ± 1.55	41.00 ± 3.49
C57Bl/6	1.90 ± 0.05 *	0.18 ± 0.01 *	59.60 ± 1.72	4.40 ± 0.49 *	87.71 ± 4.92	38.43 ± 2.10

* *p* < 0.05 compared to AY mice.

## Data Availability

The data presented in this study are available on request from the corresponding author.

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
