# Peer review of "Discovery of the First in Class 9-N-Berberine Derivative as Hypoglycemic Agent with Extra-Strong Action"

_pharmaceutics, 2021, doi:10.3390/pharmaceutics13122138_

Round 1

Reviewer 1 Report

The aim of this study was to evaluate hypoglycemic activity of novel 9-N-berberine derivate on animal model. The authors revealed that examined compound possess hypoglycemic activity due to increased insulin sensitivity. Additionally, in experimental animals (mice with T2DM and obesity) all the main markers of metabolic syndrome inherent in these mice were normalized: reduced body weight, decreased fasting glucose and insulin concentrations, and improved glucose tolerance.

The result of this study is interesting because it indicates a new compound that shows hypoglycemic effect at relatively low doses and at the same dose has a significant increasing effect on sensitivity to insulin.

The main weakness of this article is the insufficient discussion since there is practically no comparisons with articles of the similar research.

Title is suitable.

Introduction

It would be desirable to explain the reason (idea) for the choice N-berberine derivate to evaluate hypoglycemic activity.

It is necessary to give stronger explanation the aim of the work, since it has compounds with hypoglycemic effect. For example, find a berberine derivative with a lower effective dose...

Histological evaluation was performed on pancreas and brown fat pad too (lines 80-81)

Lines 68 and 71 - add a reference.

A reference 28 is unnecessary.

Materials and Methods

Indicate how many mice there are in experiment and correct weight of AY mice (according to further data they weigh more than stated)

2.2.2.-state which groups of mice were investigated

           - line 131 all compounds were mixed.. -state compounds and their quantity

           - line 132- full name of VLD tablets, dosage and function in experiment

           - state how the blood was taken (from tail?)

It needs to be explained the reason why in OGTT in C57BL/6 mice VLD tablets were used, and in OGTT in AY mice MF was used.

2.2.4. How is it single administration if experiment was conducted every other day?

Results

Figure 4. AY mice instead of blank for equal labelling

They are missing Figure 6 and 7 in the text.

Discussion

It is necessary to compare and discuss the obtained results with similar research and confirm the claim from the title that examined berberine derivative has extra strong hypoglycemic action.

The same text is listed twice (lines 344 – 357 need to be removed)

It is necessary to use full name when the term is first time used (line 132-VLD, line 142-MF...)

Author Response

Thank you for your comments!

Referee 1. Answers.

Introduction

It would be desirable to explain the reason (idea) for the choice N-berberine derivate to evaluate hypoglycemic activity.

It is necessary to give stronger explanation the aim of the work, since it has compounds with hypoglycemic effect. For example, find a berberine derivative with a lower effective dose...

Answer: It has been corrected in the revised version of article.

Histological evaluation was performed on pancreas and brown fat pad too (lines 80-81)

Answer: Thank you. Corrected.

Lines 68 and 71 - add a reference.

Answer: A reference was added.

A reference 28 is unnecessary.

Answer: In our opinion it is needed since it gives a reference to our work in which the described changes in animals were previously shown and therefore, such a model is also suitable for this study.

Materials and Methods

Indicate how many mice there are in experiment and correct weight of AY mice (according to further data they weigh more than stated)

Answer: Done.

2.2.2.-state which groups of mice were investigated

           - line 131 all compounds were mixed.. -state compounds and their quantity

           - line 132- full name of VLD tablets, dosage and function in experiment

           - state how the blood was taken (from tail?)

Answer: Thank you! Everything is corrected.

It needs to be explained the reason why in OGTT in C57BL/6 mice VLD tablets were used, and in OGTT in AY mice MF was used.

Answer: VLD as the DPP4 inhibitor increases insulin secretion [Thornberry, N.A.; Gallwitz, B. Mechanism of action of inhibitors of dipeptidyl-peptidase-4 (DPP-4). Best Pract. Res. Clin. Endocrinol. Metab. 2009, 23, 479–486.] and is routily used in our lab as a positive control in screening by OGTT since quick effect and dose rate similar (due to molecular weights) to new chemicals tested. We used metformin at a dose of 250 mg / kg as a positive control, which is a widely used first line antihyperglycemic treatment for patients with type 2 diabetes. The beneficial effects of metformin are attributed to lowering blood glucose by inhibiting hepatic gluconeogenesis and enhancing insulin sensitivity, so it was more suitable for our experiment. Such text is included into the article.

2.2.4. How is it single administration if experiment was conducted every other day?

Answer: Here was the single administration followed by blood sampling and insulin evaluation but we used same animals in different days in order to decrease inter-animal variability. We have changed slightly the text for better perception.

Results

Figure 4. AY mice instead of blank for equal labelling

Answer: Thank you! Corrected.

They are missing Figure 6 and 7 in the text.

Answer: Included into the text.

Discussion

It is necessary to compare and discuss the obtained results with similar research and confirm the claim from the title that examined berberine derivative has extra strong hypoglycemic action.

Answer: Actually such comparison is present in the introduction part in order to show readers what was done in the field but we also added some text into the discussion part, highlighted. Under term “extra strong” we mention its effect in a small dose and low LD50 due to the prominent glucose-lowering effect.

The same text is listed twice (lines 344 – 357 need to be removed)

Answer: Thank you! Removed. 

It is necessary to use full name when the term is first time used (line 132-VLD, line 142-MF...)

Answer: Corrected, highlighted

Reviewer 2 Report

This is an interesting paper regarding a potential new compound aimed at fighting diabetes. As such, it merits publication. Nevertheless, I think that some additional aspects should be discussed in the Discussion section. 

I strongly advise that a paragraph regarding potential clinical benefit of the novel drug, not just comparing it with the existing drugs, but contemplating its position in regard to mechanisms of action, as they are plenty, but the magnitude of their effect cannot be deduced from the current manuscript. Also, the authors describe those mice died of hypoglycaemic shock, that leading to an important safety concern. The authors should suggest why in spite of that, this drug should be further investigated, and how safety concerns in this regard could be addressed. Also, it would be interesting to know, whether any other side effects in the treated mice were observed and the effect of the drug on the microbiota of the mice?

Author Response

Answer: We have added some text into the discussion part. In the article we say that SHE-196 can be “first one in a novel line of therapeutic agents for diabetes mellitus treatment”, so we will search for similar compounds but with less pronounced effect in order to increase LD50. And SHE-196 can be a good one for study mechanism of action of compounds of this line. We have studied liver and pancreas histology and didn’t find any toxicity, also animals were not differed in behavior and appearance from untreated mice. We didn’t study animal’s microbiota in this experiment. 

Reviewer 3 Report

The review paper is original and potentially of interest.

I found that this paper is very interesting and that the obtained results are very promising in therapeutic agents for the treatment of diabetes mellitus.

Minor concern

[1] Figure 1 and Figure 2 should be moved to the Method section.

[2] It is not clear what is the purpose of the study, please review the title, the objective, and your material and methods section they should reflect the purpose and be integrated.

[3] The method section has a lot of subsections, the authors made wrong numbers in lines 161,167, and 171. I suggest decreasing the number of sections and following the journal format.

Author Response

Thank you for your comments! 

Referee 3. Answers.

[1] Figure 1 and Figure 2 should be moved to the Method section.

Answer: We think that it is not necessary since that figures are presenting structures of berberine derivatives that were synthesized in other works previously and the structure of our molecule and such presentation is suitable for better comparison between compounds.

[2] It is not clear what is the purpose of the study, please review the title, the objective, and your material and methods section they should reflect the purpose and be integrated.

Answer: The purpose of the study was rewritten and placed at the end of the introduction part.

[3] The method section has a lot of subsections, the authors made wrong numbers in lines 161,167, and 171. I suggest decreasing the number of sections and following the journal format.

Answer: Subsections are needed for better perception of different methods used in our work. Similar style was used in already published articles, for example: https://www.mdpi.com/1999-4923/13/10/1739/htm, https://www.mdpi.com/1999-4923/13/10/1736/htm, https://www.mdpi.com/1999-4923/13/11/1979/htm. Numbering sequence corrected.